# Immunity Debt Regarding the Aspect of Influenza in the Post-COVID-19 Era in Taiwan

**DOI:** 10.3390/v16091468

**Published:** 2024-09-15

**Authors:** Edward Wu, Victoria Wu, Kang-Hsi Wu, Kun-Chan Wu, Jing-Yang Huang

**Affiliations:** 1Morrison Academy, Taichung 406, Taiwan; wued25@ma.org.tw (E.W.); wuv27@ma.org.tw (V.W.); 2Department of Pediatrics, Chung Shan Medical University Hospital, Taichung 402, Taiwan; cshy1903@gmail.com; 3School of Medicine, Chung Shan Medical University, Taichung 402, Taiwan; 4Wu Kun Chan Clinic, Miaoli 360, Taiwan; wupedhanwupedhan@gmail.com; 5Institute of Medicine, College of Medicine, Chung Shan Medical University, Taichung 402, Taiwan; 6Department of Medical Research, Chung Shan Medical University Hospital, Taichung 402, Taiwan

**Keywords:** COVID-19, influenza, hospitalization, Taiwan, immunity debt

## Abstract

Immunity debt for various viral infections was reported globally in the post-COVID-19 era, but the data about influenza are lacking. This study collected data from Taiwan’s CDC Open Data Portal. We analyzed the weekly number of influenza hospitalizations from January 2017 to May 2024. We divided the study period into four phases: the pre-COVID-19 without influenza epidemics, pre-COVID-19 with an influenza epidemic, COVID-19 pandemic lockdown control, and COVID-19 pandemic unlock periods. The Wilcoxon rank-sum test and interrupted time series analysis were used. The median case numbers of the four time periods were 174 (IQR = 98), 431 (IQR = 160), 8, and 155 (IQR = 175), respectively. Under the COVID-19 pandemic lockdown control, the weekly influenza hospitalization case number decreased by 90.2% (*p* < 0.001). The non-pharmaceutical intervention (NPI) policies during the COVID-19 pandemic helped Taiwan reduce influenza hospitalizations significantly. Till now, a comparison of the prevalence of influenza pre-COVID-19 and post-COVID-19 has yet to be reported. In our study, with the pandemic unlocking, it increased by 20-fold (*p* < 0.001), but the case number was still significantly lower than that pre-COVID-19. Amongst other factors, this may be associated with continuing self-induced NPIs in Taiwan.

## 1. Introduction

The SARS-CoV-2 virus, which causes COVID-19 in humans, first emerged in December 2019, and the virus spread quickly, which severely impacted countries worldwide. In response to the COVID-19 crisis, many nations implemented various measures, such as quarantine, isolation, border controls, travel restrictions, and other non-pharmaceutical interventions (NPIs) [1]. Numerous experiences and studies have, indeed, proven that various NPIs can effectively reduce the spread of both COVID-19 and influenza [1,2,3]. However, this does not just apply to influenza. During the global COVID-19 pandemic, the implementation of NPIs led to a noticeable drop in the transmission of respiratory infectious diseases, such as respiratory syncytial virus (RSV), common human coronaviruses, parainfluenza viruses, human metapneumovirus, adenovirus, and rhinovirus [4,5,6,7]. After a little more than three years, the COVID-19 pandemic showed signs of easing. Along with the evolution of SARS-CoV-2 variants and the herd immunity formed from the vast majority of people who had previously been infected with COVID-19, countries around the world began to start lifting restrictions around the years 2022–2023 [1]. As pandemic prevention measures relaxed, countries around the world observed a notable increase in cases of respiratory virus infections, a phenomenon known as “immunity debt” [1,2,8,9,10,11,12,13,14,15,16,17]. These viruses include well-known ones such as RSV, parainfluenza, and seasonal coronavirus and many other viral infections, such as rhinoviruses, enteroviruses, metapneumovirus, and adenoviruses [1,2,8,9,10,11,12,13,14,15,16,17].

During the period of global effort to contain COVID-19, Taiwan emerged as a model for pandemic prevention. As an island nation with a population of approximately 23 million and no direct land borders with other countries, Taiwan implemented “immediate”, “rapid”, and “proactive” preventive policies, which included preventing visitors from the epidemic area from entering the country, quarantining incoming travelers, conducting contact tracing, isolating confirmed patients, enforcing mask mandates, and preventing large gatherings. These comprehensive and aggressive measures quickly reduced the number of acute COVID-19 infections in Taiwan, bringing the situation under control [18,19]. During Taiwan’s COVID-19 epidemic-containing period, a rapid decline in emergency department visits for respiratory viruses such as influenza and enterovirus was observed. Critical cases of influenza decreased by 95–98% during this time [1,20]. Following the emergence of COVID-19 vaccines, Taiwan’s population also widely received COVID-19 vaccinations. As of 1 March 2023, among Taiwan’s 23 million people, a total of 67.12 million doses of COVID-19 vaccines had been administered [21]. Eventually, Taiwan successfully contained COVID-19 over the course of three more years. By the winter of 2022, with the overall easing of the COVID-19 pandemic, Taiwan began gradually implementing reopening policies [22]. Not surprisingly, right after the relaxation of the NPI policies, Taiwan’s CDC data showed an increase in respiratory virus positivity rates [23].

Despite the occurrence of several immunity debts related to respiratory viruses worldwide and in Taiwan, there remains relatively little discussion of immunity debt specifically related to influenza. Examples include one of China’s “forecasts” predicting that influenza “will likely increase” after China’s reopening by using a mathematical model. It is also forecasted that if NPIs are relaxed after achieving the zero policy, influenza cases in China could significantly increase in 2022–2023 [24]. In advance, a study of Romanian hospitalized children reported an increase in positivity rates of rapid influenza antigen tests after relaxing restrictions for COVID-19 [25]. A report comparing the USA and England found that “The positive rate of influenza has a brief surge after relaxing NPIs” [8]. Data from China indicate “There was an immunity debt of influenza lasting for about two months evidenced by increased influenza positive rate after the pandemic” [9]. All the articles above reveal that the trend of influenza infection during the post-COVID-19 era may have a pattern of see-sawing or upsurging. These studies focused on comparing the rebound of influenza during the pandemic control periods versus after relaxing. Many COVID-19 hospitalization and influenza hospitalization surveillance databases, such as FLUVIEW, NHSN, RESP-NET, the National Influenza and COVID-19 Surveillance Report from the UK, and the COVID-19 Epidemiological Update from the WHO, do not seem to have data comparing influenza hospitalizations from before and after COVID-19 [26,27,28,29,30].

Lastly, the WHO’s FluNet shows a significant drop in global “influenza specimen numbers” during the pandemic period, with noticeably higher numbers in both the pre-pandemic and post-pandemic periods. Unfortunately, the data only present the “positivity rate of influenza specimens” before and after the pandemic, without indicating the overall trend in the “case numbers” of influenza cases [14,31]. Until now, there have not been many real-world comparisons specifically addressing how the prevalence of influenza has changed before versus after COVID-19 prevention measures. As such, it remains unclear whether influenza cases truly rose after the pandemic reopening or exceeded pre-COVID-19 prevention levels. Taking Taiwan’s experience as a starting point, this investigation sought to explore the actual trends in hospitalizations from influenza following the lifting of COVID-19 prevention measures. Interestingly, a unique trend of influenza hospitalizations in Taiwan’s post-COVID-19 era was found.

## 2. Materials and Methods

### 2.1. Data Source

This study collected data from publicly available information in Taiwan’s CDC Open Data Portal [32]. The weekly number of cases of influenza hospitalizations in Taiwan was used for analysis [33]. In addition, the case numbers of influenza viral-associated pneumonia were also included. The data were collected from Taiwan’s National Health Insurance (NHI) database, specifically the cases of hospitalizations from influenza and influenza viral-associated pneumonia (VAP) for ICD-10 diagnostic codes J09.x, J10.x, and J11.x. The data were compiled on a weekly basis, allowing the NHI to have comprehensive weekly statistical information, which was representative of the Taiwanese population of 23 million. Due to the 99.9% coverage rate of Taiwan’s universal health coverage [34], these data were nationally representative of Taiwan for the period from 2017 to May 2024.

### 2.2. Periods of COVID-19 Pandemic Lockdown Control and Influenza Pandemic

We collected data on the number of influenza hospitalizations from January 2017 to May 2024. The date of the first imported COVID-19 case in Taiwan was 21 January 2020. However, the Taiwan CDC announced the activation of the Central Epidemic Command Center (CECC) for Severe Special Infectious Pneumonia on 20 January 2020 [22], which is indicated by the red solid line in Figure 1. On 13 October 2022, Taiwan completely lifted the mandatory quarantine requirements for inbound travelers, thus gradually easing the COVID-19 pandemic lockdown control [22], which is indicated by the green dashed line in Figure 1.

We defined the influenza epidemic threshold (indicated by the gray horizontal line; value = 356 in the figure) following the Vega et al. method and using the R ‘mem’ package for the threshold calculation [35]. According to the COVID-19 pandemic lockdown control and influenza epidemics from 2017 to May 2024, we divided the study time into four periods: pre-COVID-19 without an influenza epidemic (approximately 105 weeks), pre-COVID-19 with an influenza epidemic (approximately 55 weeks), COVID-19 pandemic lockdown control (approximately 141 weeks), and COVID-19 pandemic unlock (approximately 79 weeks). The pre-COVID-19 without an influenza epidemic and pre-COVID-19 with an influenza epidemic periods both spanned from 2017-W01 to 2020-W03. However, the pre-COVID-19 with an influenza epidemic period included five distinct influenza epidemic waves: the first epidemic from 2017-W21 to 2017-W32, the second from 2017-W52 to 2018-W11, the third from 2019-W02 to 2019-W09, the fourth from 2019-W22 to 2019-W38, and the fifth from 2019-W49 to 2020-W03. In contrast, the pre-COVID-19 without an influenza epidemic period excluded the weeks corresponding to these five distinct epidemic waves. The third period covered 2020-W04–2022-W41, and the fourth period spanned from 2022-W42 to 2024-W16 (Figure 1).

### 2.3. Statistical Analysis

The median (interquartile range (IQR)) was used to describe the distribution of weekly hospitalization cases with influenza in the four study periods. The Wilcoxon rank-sum test was used to compare the median of weekly hospitalization cases with influenza across the four study periods. We used interrupted time series analysis to analyze the following [36]: 1. the weekly increase in the proportion of influenza hospitalizations during the pre-COVID-19 period (2017-W01–2020-W03) compared with the COVID-19 lockdown period (2020-W04–2022-W41); 2. the weekly increase in the proportion of influenza hospitalizations during the COVID-19 lockdown period (2020-W04–2022-W41) compared with the COVID-19 pandemic unlock period (2022-W42–2024-W16). All statistical analyses were performed using the SAS 9.4 software, with the significance level set at 0.05 and all tests being two-tailed.

## 3. Results

### 3.1. Weekly Influenza Hospitalizations in Four Study Periods

Table 1 shows that during the pre-COVID-19 weeks not in an influenza epidemic period, the weekly hospitalization cases ranged from 53 to 385, with a median of 174 (IQR = 98) and an arithmetic mean of 186.1 (SD = 78.0). During the pre-COVID-19 weeks during the influenza epidemic periods, the weekly hospitalization cases ranged from 293 to 948, with a median of 431 (IQR = 160) and an arithmetic mean of 486.1 (SD = 150.0). In the COVID-19 pandemic lockdown control period, the weekly hospitalization cases ranged from 0 to 430, with a median of 8 (IQR = 11) and an arithmetic mean of 16.6 (SD = 44.1). In the COVID-19 pandemic unlock period, the weekly hospitalization cases ranged from 15 to 311, with a median of 155 (IQR = 175) and an arithmetic mean of 143.0 (SD = 83.6).

Figure 2 presents the box plot of weekly influenza hospitalization cases across the different periods. Compared with the median of weekly hospitalization cases during the pre-COVID-19 without an influenza epidemic period, the median during the pre-COVID-19 with an influenza epidemic period was significantly higher (*p* < 0.001). In contrast, the medians during the COVID-19 pandemic lockdown control (*p* < 0.001) and COVID-19 pandemic unlock periods (*p* = 0.004) were significantly lower. Additionally, the median of weekly hospitalization cases in both the COVID-19 pandemic lockdown control period (*p* < 0.001) and the COVID-19 pandemic unlock period (*p* < 0.001) were significantly lower than the median in the pre-COVID-19 with an influenza epidemic period. A significantly higher median of weekly hospitalization cases was observed in the COVID-19 pandemic unlock period when compared with the COVID-19 pandemic lockdown control period (*p* < 0.001). Even though there was a significant increase in influenza hospitalizations after the COVID-19 pandemic, it was still less compared with the pre-COVID-19 periods with or without an influenza epidemic, which is shown in Figure 1 and Figure 2 and Table 1.

### 3.2. Interrupted Time Series Analysis for Effect of COVID-19 Pandemic Lockdown Control Policies

Table 2 shows the interrupted time series analysis for weekly influenza hospitalization cases affected by COVID-19 pandemic lockdown control policies. The exp(β) of the intercept during the pre-COVID-19 period indicates that the initial number of weekly influenza hospitalizations was 174.3 (95% CI = 99.4–305.5). The exp(β) for the time effect during this period was 1.004 (p = 0.0963), suggesting no significant secular time trend in weekly influenza hospitalizations. During the COVID-19 pandemic lockdown period, the exp(β) for the effect between the pre-COVID-19 period and the lockdown period was 0.098 (95% CI = 0.046–0.206), indicating a 90.2% decrease in weekly influenza hospitalizations following the implementation of lockdown controls. Additionally, the exp(β) for the time effect during the lockdown period was 0.979 (95% CI = 0.970–0.989), representing a significant 2.1% weekly reduction in influenza hospitalizations during the lockdown period.

Though some papers mentioned “immunity debt” after COVID-19, it continues to be a topic worth discussing. We suppose that immunity debt is still speculative and theoretical. Whether the resurgence of influenza post-COVID-19 was associated with immunity or just a new ability of the virus to spread is still unclear. Control policies for weekly influenza hospitalization cases are needed.

### 3.3. Interrupted Time Series Analysis for the Effect of COVID-19 Pandemic Unlock Policies

Table 3 shows the interrupted time series analysis for the weekly influenza hospitalization cases affected by COVID-19 pandemic unlock policies. The exp(β) of the intercept during the lockdown period indicates that the initial number of weekly influenza hospitalizations was 894.1 (95% CI = 93.0–8599.9), consistent with the results in Table 2. During the COVID-19 pandemic unlock period, the exp(β) for the effect between the lockdown and unlock periods was 20.205 (95% CI = 7.310–55.847), indicating a 20-fold increase in weekly influenza hospitalizations following the implementation of unlock policies (*p* < 0.001). Furthermore, the exp(β) for the time effect during the unlock period was 1.031 (95% CI = 1.022–1.040), reflecting a significant 3.1% weekly increase in influenza hospitalizations during the unlock period.

## 4. Discussion

Common reasons for experts to assume the possibility of various virus immunity debts after the COVID-19 pandemic may include children and adults having prolonged periods without exposure to viral infections, resulting in insufficient serum antibody levels to combat the viruses [37]. Another consequence of more than three years without infection is that some newborns cannot receive antibodies from their mothers, and there are also reports that COVID-19 infection can compromise the immune system [9]. Though some papers mentioned “immunity debt” after COVID-19, it continues to be a topic worth discussing. We suppose that immunity debt is still speculative and theoretical. Whether the resurgence of influenza post-COVID-19 is associated with immunity or just a new ability of the virus to spread is still unclear.

There are not many studies specifically discussing the influenza immunity debt after the COVID-19 pandemic, and they are almost all lacking long-term tracking. The data in this study, conversely, covered a span of 7.5 years including the COVID-19 pandemic period, pre-pandemic period, and post-pandemic period (from January 2017 to May 2024). They explored whether the fluctuation in the number of influenza hospitalizations in Taiwan during this period was associated with the NPI policies implemented for COVID-19 prevention. We chose to study the cases from influenza hospitalizations rather than non-hospitalization cases because all the data on the case numbers came exactly from the reported hospitalization results.

It is encouraging that after implementing extensive lockdowns and various NPIs, Taiwan experienced a successfully contained status for both COVID-19 and influenza for over three years. After the lifting of the NPIs, Taiwan did experience a rebound in terms of influenza hospitalizations, but the rebound was not extreme (Figure 1 and Figure 2; Table 1). Even though the case number of hospitalizations for influenza did increase abruptly following the lifting of mandatory NPIs for COVID-19, the increase was characterized as a low-level epidemic. However, this increase occurred because, during the COVID-19 epidemic period, the number of hospitalization cases for influenza was extremely low and started from a low baseline. When compared with the hospitalization numbers before the COVID-19 epidemic prevention measures, the hospitalization numbers for influenza were lower (*p* < 0.0001; Figure 2). They were even lower than the level of the low-influenza season before the COVID-19 pandemic (*p* = 0.0043; Figure 2). Taiwan did experience a moderate immunity debt for influenza hospitalizations post-COVID-19 control measures. The observation that the number of influenza hospitalizations immediately decreased after the COVID-19 control period was strongly associated with the effectiveness of Taiwan’s strict COVID-19 pandemic control (Figure 1; Table 2). The immediate rebound of influenza hospitalizations after the relaxation of NPIs (Figure 1; Table 3), again, indicates that good NPI policies suppressed influenza virus infections.

But why was there not any strong and obvious immunity debt for influenza hospitalizations after Taiwan lifted various NPIs for COVID-19? A good explanation is that after more than three years of COVID-19 pandemic measures in Taiwan, the population had become accustomed to NPIs, including the habit of wearing masks diligently. Even after the government lifted the mask mandate, many people continued to voluntarily wear masks and adhere to NPIs. Additionally, Taiwan has a consistent policy of annual influenza vaccinations, which continued during the COVID-19 pandemic. In Taiwan, influenza vaccinations are provided annually to vulnerable groups, such as the elderly, young children, and immunocompromised patients. Since 2016–2017, the number of free vaccines administered each year has been as follows: 6024.3K, 5992.8K, 5314.4K, 5997.0K, 6230.3K, and 6065.2K. It is apparent that the number of people receiving flu vaccines did not decrease during and after the COVID-19 pandemic but slightly increased [38]. However, despite these measures, there are still various respiratory viral infections after the lifting of restrictions worldwide and in Taiwan. If multiple outbreaks occur simultaneously, the consequences of multi-respiratory virus immunity debt could still lead to a sudden increase in healthcare burdens, resulting in a potential decrease in the quality of patient care. Fewer viral illnesses result in fewer secondary bacterial infections [1,16,32]. We contend that it is a better option to take a gradual approach when lifting NPIs. We must remain vigilant against the potential rebound of immunity debt for various viruses after any pandemic control measure.

Overall, there was no overload of hospital wards and intensive care units during the post-COVID-19 era in Taiwan. From this perspective, the Taiwan model has, indeed, demonstrated commendable achievements. Facing any potential new pandemics in the future, we highly recommend Taiwan’s COVID-19 prevention model. This involves initially implementing aggressive NPIs to enforce strict containment measures, decisively interrupting transmission pathways to reduce the burden on healthcare systems. This approach predates vaccine availability, anticipates virus attenuation, and expects epidemic attenuation before proceeding with reopening. The gradual lifting of NPIs and active promotion of influenza vaccination follow suit. This model is estimated to offer an effective strategy for controlling new emergent respiratory infection epidemics or pandemics in the future without causing an excessive outbreak of influenza post-pandemic.

However, our study had limitations. Our data were from the CDC database; therefore, we were unable to differentiate influenza cases based on virus diversity or strain variations due to influenza viruses’ propensity for antigenic drift or shift. This makes their epidemiology less predictable. We were also unable to account for the degree of match or mismatch between circulating influenza strains and details of which vaccines are deployed at certain ages and in certain risk groups. Moreover, even though we found that the number of influenza hospitalizations did rebound in the post-COVID-19 era in Taiwan and the level of increase was not so high, there is still a lack of global data. Certainly, tracking the longer-term influenza trends following the COVID-19 pandemic is also worth promoting.

## 5. Conclusions

Influenza hospitalizations were controlled with NPIs and, of course, returned when the lockdown ended in Taiwan. However, the Taiwan model of COVID-19 prevention resulted in influenza hospitalizations not experiencing an excessive rebound compared with the pre-COVID-19 period.

## Figures and Tables

**Figure 1 viruses-16-01468-f001:**
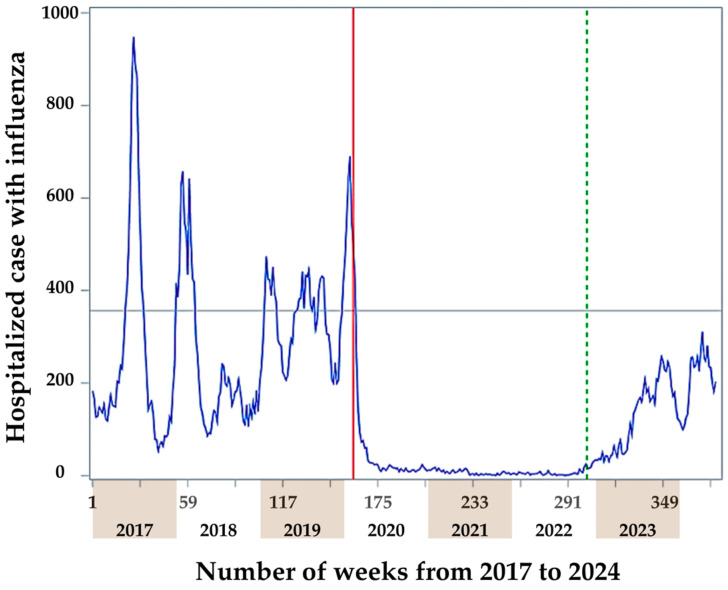
Weekly hospitalization cases with influenza and COVID-19 pandemic lockdown control. The blue line indicates the weekly number of influenza hospitalizations. The red solid line indicates the Taiwan CDC announced the activation of the Central Epidemic Command Center for Severe Special Infectious Pneumonia on 20 January 2020. The green dashed line indicates Taiwan completely lifted the mandatory quarantine requirements for inbound travelers on 13 October 2022. The gray horizontal line indicates the influenza epidemic threshold; the value is 356.

**Figure 2 viruses-16-01468-f002:**
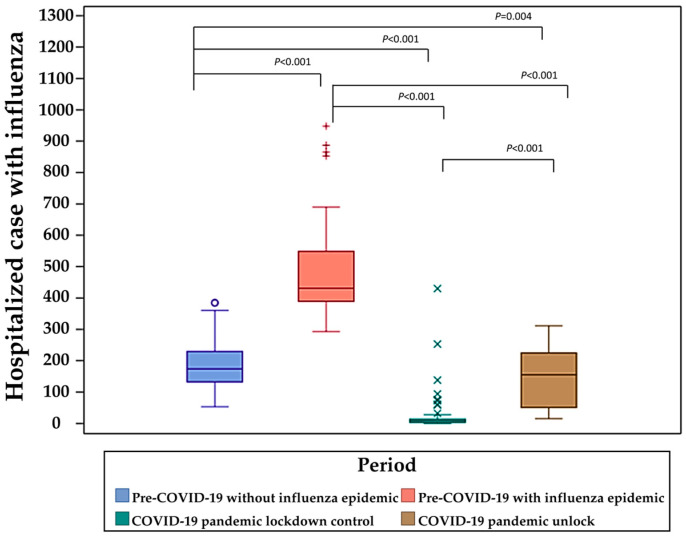
Comparison of influenza hospitalization cases across different periods. The box plot of weekly influenza hospitalization cases across the study periods. The *p*-values indicated are from the Wilcoxon rank-sum test used to compare the median of weekly influenza hospitalizations between the different periods.

**Table 1 viruses-16-01468-t001:** Weekly influenza hospitalization cases stratified by pre-COVID-19 influenza epidemic periods and COVID-19 pandemic lockdown control policies.

		Weekly Influenza Hospitalization Cases		
Period	No. of Weeks	Min–Max	Median (IQR)	Arithmetic Mean (SD)
Pre-COVID-19 without influenza epidemic	105	53 to 385	174 (98)	186.1 (78.0)
Pre-COVID-19 with influenza epidemic	55	293 to 948	431 (160)	486.1 (150.0)
COVID-19 pandemic lockdown control	141	0 to 430	8 (11)	16.6 (44.1)
COVID-19 pandemic unlock	79	15 to 311	155 (175)	143.0 (83.6)

**Table 2 viruses-16-01468-t002:** Interrupted time series analysis for the effect of COVID-19 pandemic lockdown control policies on weekly influenza hospitalization cases.

Parameter	exp(β)	95% CI of exp(β)	*p*-Value
Intercept in pre-COVID-19 period	174.3	99.4–305.5	<0.001
Time effect in pre-COVID-19 period	1.004	0.999–1.009	0.096
Effect between pre-COVID-19 period and COVID-19 pandemic lockdown control period	0.098	0.046–0.206	<0.001
Time effect in COVID-19 pandemic lockdown control period	0.979	0.970–0.989	<0.0001

**Table 3 viruses-16-01468-t003:** Interrupted time series analysis for the effect of COVID-19 pandemic unlock policies on weekly influenza hospitalization cases.

Parameter	exp(β)	95% CI of exp (β)	*p*-Value
Intercept in COVID-19 pandemic lockdown control period	894.1	93.0–8599.9	<0.001
Time effect in COVID-19 pandemic lockdown control period	0.979	0.970–0.989	<0.001
Effect between COVID-19 pandemic lockdown control period and COVID-19 pandemic unlock period	20.205	7.310–55.847	<0.001
Time effect in COVID-19 pandemic unlock period	1.031	1.022–1.040	<0.001

## Data Availability

The data on influenza hospitalizations used in this study are publicly available from the Taiwan Centers for Disease Control (CDC) website. The weekly number of influenza hospitalizations can be accessed at http://www.cdc.gov.tw/. These data are open access and do not require special permission for use.

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
