# Peer review of "Immunity Debt Regarding the Aspect of Influenza in the Post-COVID-19 Era in Taiwan"

_viruses, 2024, doi:10.3390/v16091468_

Round 1

Reviewer 1 Report

Comments and Suggestions for Authors

This is a very interesting examination of influenza hospitalization rates before and after the Covid 19 pandemic lockdown in Taiwan, demonstrating clearly the incredible success of the non pharmaceutical interventions (NPIs) to stop the spread of respiratory viruses including influenza.  Physicians around the globe observed this phenomenon in their practices but you have carefully documented it, and this is a service to public health. 

Whether this can be ascribed to "immunity debt" is unclear. It is possible that influenza could not pass from person to person during the lockdown, and returned in its normal way following the end of lockdown.  

A second concern is, did you wait long enough post covid to see return of influenza to pre lockdown rates.  You do  mention in discussion that the lower rates might be attributable to many people continuing precautions such as masking.  

Specific comments:
1. I recommend using the term "immunity debt" as a speculative or theoretical phenomenon.

2. Introduction line 88: the word "increase" is not appropriate for something in the past. Instead say something like: "Higher numbers in the pre and post..."

3. Methods, line 117: "Until October...": Do you mean "On October..."

4. Lines 128-134: I was confused by this until I figured out what was meant by "without influenza epidemic".  I recommend you take a separate sentence to explain that you have counted how many weeks there were during the epidemic periods, and how many weeks between epidemic periods. Then when you have defined or explained this, in a separate sentence provide the numbers of weeks. 

5. Results: line 156.  Reword, for example instead of "without epidemic period", say "weeks not in an epidemic period."

6. Similarly, line 158, say "weeks during epidemic periods...".

7. Line 193, 196 (and elsewhere): Please define "increase rate" and "decrease rate" or reword those.  None of the tables uses the term "increase rate".  Do you mean "rate" or incidence? This is unclear to me.  

8. Discussion lines 224-226: I understand you are saying that the data are reliable because all cases of hospitalization due to influenza are counted; which is not the case for non hospitalized cases.  The way you have worded this is unclear, please rephrase the sentence. 

9. Line 231: I do not think you have proved that the return of influenza is due to "immunity debt".  Please consider using immunity debt as a theory and  not necessarily the reason for return of influenza. 

10. Line 238: "Numbers for influenza are not actually higher": do you mean, the numbers are lower? If so that would be a more direct way of saying this. 

11. Line 262: Why mention antibiotics? These would not be used for most viruses; perhaps you mean, fewer viral illnesses and thus fewer secondary bacterial infections. 

12. Line 275; You mean respiratory viral infection epidemics.  (Other types of viral epidemics could not be controlled in the same way)

13. Line 279: Does influenza actually mutate? I know people use the term "Shift" and "drift".  This is my uncertainty about the terms.

14. Line 285: I would say, influenza was controlled with the NPIs, and of course returned when lockdown ended.  Whether this is "immunity debt" or just new ability of the virus to spread is not clear. 

Your data are great, some clarification in language is recommended. 

Comments on the Quality of English Language

Great data, some need for rewriting, clarification, as noted above.

Author Response

Dear Reviewer 1, 
Thank you for thoroughly reviewing our paper. We have revised our manuscript and
added the information as you suggested.
I agree with your opinion on immunity debt, so we have summarized your comments
and added it to our paper as mentioned below: Though some papers mentioned
“immunity debt” after COVID-19, it continues to be a topic worth discussing. We
suppose that immunity debt is still speculative and theoretical. Whether the

resurgence of influenza post-COVID-19 is associated with immunity or just a new
ability of the virus to spread is still unclear. 

1. I recommend using the term "immunity debt" as a speculative or theoretical
phenomenon.
Reply: We agree with your opinion about “immunity debt”. We’ve summarized your
ideas and have added them to our paper. 

2. Introduction line 88: the word "increase" is not appropriate for something in the
past. Instead say something like: "Higher numbers in the pre and post..."
Reply: We changed the word “increase” to “noticeably higher numbers”

3. Methods, line 117: "Until October...": Do you mean "On October..."
Reply: We have changed it from "Until October..." to “On October…”

4. Lines 128-134: I was confused by this until I figured out what was meant by
"without influenza epidemic".  I recommend you take a separate sentence to explain
that you have counted how many weeks there were during the epidemic periods, and
how many weeks between epidemic periods. Then when you have defined or
explained this, in a separate sentence provide the numbers of weeks. 
Reply: We have taken a separate sentence to define explain that we have counted how
many weeks there were during the epidemic periods and how many weeks between
epidemic periods. We have also provided the numbers of weeks in a separate
sentence. 

5. Results: line 156.  Reword, for example instead of "without epidemic period",
say "weeks not in an epidemic period."
Reply: We have changed the text from “without epidemic period” to “weeks not in an
influenza epidemic period”. 

6. Similarly, line 158, say "weeks during epidemic periods..."
Reply: We have changed the text to “weeks during influenza epidemic periods”
instead of “with influenza epidemic period” to clarify things. 

7.  Line 193, 196 (and elsewhere): Please define "increase rate" and "decrease rate"
or reword those.  None of the tables uses the term "increase rate".  Do you mean
"rate" or incidence? This is unclear to me.
Reply: We used the phrases increase rate because we are measuring the average
increase and decrease of the number of cases, not the individual cases. However, we
understand the confusion, and we have reworded the paragraph so that what we are
conveying is more clear. 

8. Discussion lines 224-226: I understand you are saying that the data are reliable
because all cases of hospitalization due to influenza are counted; which is not the case
for non hospitalized cases.  The way you have worded this is unclear, please rephrase
the sentence. 
Reply: We have reworded the sentence to “We chose to study the cases from
influenza hospitalizations rather than non hospitalization cases because all the data on
the case numbers come exactly from the reported hospitalization results”. We chose to
use hospitalization case numbers because the hospitalization case numbers are more
reliable. 

9. Line 231: I do not think you have proved that the return of influenza is due to
"immunity debt".  Please consider using immunity debt as a theory and not
necessarily the reason for return of influenza. 
Reply: We agree that immunity debt as a theory and is not the reason for the return of
influenza. We have summarized ideas of immunity debt and put it in a paragraph.

10. Line 238: "Numbers for influenza are not actually higher": do you mean, the
numbers are lower? If so that would be a more direct way of saying this. 
Reply: Yes, we have now changed it so that it says “lower”; that makes it clearer. 

11. Line 262: Why mention antibiotics? These would not be used for most viruses;
perhaps you mean, fewer viral illnesses and thus fewer secondary bacterial infections. 
Reply: Agreed, we have modified our sentence so that in conveys clearly that fewer
viral illnesses results in fewer secondary bacterial infections

12. Line 275; You mean respiratory viral infection epidemics.  (Other types of viral
epidemics could not be controlled in the same way)
Reply: Yes, we have changed it to respiratory.

13.  Line 279: Does influenza actually mutate? I know people use the term "Shift"
and "drift".  This is my uncertainty about the terms.
Reply: We have clarified that it is not a mutation but rather an antigenic drift or shift. 

14. Line 285: I would say, influenza was controlled with the NPIs, and of course
returned when lockdown ended.  Whether this is "immunity debt" or just new ability
of the virus to spread is not clear. 
Reply: We agree. We have changed our conclusion so that immunity debt is a possible
explanation but we are unsure if that is the actual reason. We have removed the
sentence that is unclear. 

Thanks again for your valuable comments. 

Sincerely, 
Edward Wu
Email: wued25@ma.org.tw

Reviewer 2 Report

Comments and Suggestions for Authors

Thank you for this interesting paper. I would suggest resubmission once you have addressed the concerns I will outline below and the correction of some of the language quality I will further detail.

My main concern about this paper is whilst you quantify the magnitude of the changes in your findings you overstate in the summary, discussion and conclusions a causal relationship when other possible explanations cannot be excluded nor are they mentioned. I would request that whilst some of this can be addressed in an expansion of the limitations section that you need to amend the text to say your findings "are consistent with" or "are associated with" rather than make too sweeping categorical statements.

In your consideration of what needs to be addressed in the summary, discussion and conclusion some of this may be due to a translation issue but the strength of the conclusion are overstated given the absence of virological data for comparison of the degree of match/mismatch between circulating influenza strains and details of which vaccines are deployed/which age/risk groups are affected c.f. which age groups/risk groups are vaccinated. The former is important since a good match between circulating virus and that in the vaccine in the most recent season could account a reduced impact on hospitalisation. For the latter better targeting of or uptake in those most at risk could result in reduced hospitalisation.  

You and your co-authors need to soften the categorical language in the summary, discussion and conclusion and reflect these further limitations as suggested in the response to the authors before resubmission.

I would request that you make reference to openly available surveillance data from other countries which is published on hospitalisation trends for influenza and COVID-19. Such data demonstrates country specific patterns globally and you could entreat that applying the methodology you have used that other investigators could assess the changes they have observed.

I had to do a bit of mental gymnastics to convert what your authors have found in their tables into the text. I think this can be overcome by insertion of a sentence or two to assist this process - in the text you present a set of figures which are e.g. 100 - the finding in the table etc.

Comments on the Quality of English Language

I would suggest amendment to the language used as follows;

Line 26/27 -  ", which may be associated with self-induced NPIs in Taiwan." I would suggest amending to "Amongst other factors this may be associated with continuing self-induced NPI's in Taiwan"

Line 44 - "Along with the evolution of SARS-CoV-2 variants and the herd immunity formed from the vast majority of people that had previously get COVID-19" replace the word "get" - perhaps with the words "been infected with"

Line 47-48 - "countries around the world observed a notable increase in cases of respiratory virus infections, a phenomenon known as immunity debt," I would suggest placing inverted comma's at "immunity debt" and insert the same references that you then place at the end of the paragraph.

Line 75-76 - "In advance, research of a Romanian children’s hospital says “The positive rate of rapid influenza antigen test increased after relaxing restrictions for COVID-19” [25]. I would suggest amending to "A study in Romanian hospitalised children reported an increase in positivity rates of rapid..."

Line 77 - "A report comparing the USA and England found out that “ delete the word "out".

Line 82-83 - "But the focuses of them all are on comparing the rebound of influenza during pandemic control periods versus after relaxing." Amend to "These studies focus on comparing..."

Line 179 - remove "it is interesting that"

Line 222 - "It explores whether the number of influenza hospitalizations in Taiwan during this period fluctuated due to the NPIs policies implemented for COVID-19 prevention. " I would suggest changing to "It explores whether the fluctuation in the number of influenza hospitalizations in Taiwan during this period was associated with NPIs policies implemented for COVID-19 prevention."

Line 223 - Amend to "chose" rather than "choose".

Line 228 - "This study proves that Taiwan, after implementing extensive lockdowns and various NPIs, had successfully contained both COVID-19 and influenza over three years". I would suggest amending this categorical statement to "It is encouraging that after implementing extensive lockdowns and various NPIs which had successfully contained both COVID-19 and influenza, that Taiwan has thus far experience a limited  in which had successfully contained both COVID-19 and influenza over three years, 

Line 234 - Remove the word "that" after "level"

Line 241 - "The fact that the number of influenza hospitalizations immediately decreased after COVID-19 control proved the effectiveness of Taiwan’s strict COVID-19 pandemic control (Fig.1, Table 2).  Replace the word "fact" with the word "observation". I would suggest replacing the word "proved" with "is strongly associated with" or "very likely attributable" or a similar phrase.

Line 258 - "However, despite these measures, there are still various respiratory viral infections after the lifting of restrictions worldwide and Taiwan." Please insert the word "in" before the word "Taiwan".

Line 262 - "This is why we think it’s a better option to take a gradual approach when lifting NPIs." I suggest amending to "We contend that it is a better..."

Line 271 - " This approach waits for vaccine availability, anticipates virus attenuation, and expects epidemic attenuation before proceeding with reopening." I would suggest replacing the words "waits" with "predates".

Author Response

Dear Reviewer 2,
Thank you for thoroughly reviewing our paper. We gratefully acknowledge your
comments and suggestions, which are valuable in improving the quality of our
manuscript. We revise our manuscript accordingly and add your comments to this
article. We hope that our revisions meet your expectations.

We understand we might be jumping to conclusions, so we have made modifications
to our manuscript to not overstate. There are some limitations in this manuscript: we
are unable to differentiate influenza cases based on virus strains or diversity and we
are unable to account for the degree of match or mismatch between circulating
influenza strains and details of which vaccines are deployed at certain ages and risk
groups. These limitations are reflected in the manuscript.

We have softened our descriptions in the summary, discussion, and conclusion. We
also have reflected on our limitations in this manuscript. We have made references to
openly available surveillance data from other countries in our introductions and
references.

We have made additions to our paragraph that had made the connection between the
tables and the text for better reading and understanding.

We have made changes of the quality of English language to our paper according to
your valued suggestions. 

Thank you again for your valuable insights.

Sincerely, 
Edward Wu
Email: wued25@ma.org.tw